# Role and knowledge of nurses in the management of non-communicable diseases in Africa: A scoping review

Jean Toniolo[1,2,3]*, Edgard Brice Ngoungou[1,2,4], Pierre-Marie Preux[1], Pascale Beloni[1,3]

**1** Inserm U1094, IRD UMR270, Univ. Limoges, CHU Limoges, EpiMaCT - Epidemiology of Chronic Diseases in Tropical Zone, Institute of Epidemiology and Tropical Neurology, OmegaHealth, Limoges, France, **2** Département d'Epidémiologie Biostatistiques et Informatique Médicale (DEBIM)/ Unité de Recherche en Epidémiologie des Maladies Chroniques et Santé Environnement (UREMCSE), Faculté de Médecine, Université des Sciences de la Santé, Owendo, Gabon, **3** Département Universitaire de Sciences Infirmières, Faculté de Médecine et Pharmacie, Université de Limoges, Limoges, France, **4** Centre d'Epidémiologie, de Biostatistique, et de Méthodologie de la Recherche-Gabon (CEBIMER-Gabon), Institut Supérieur de Biologie Médicale (ISBM), Université des Sciences de la Santé, Owendo, Gabon

* jean.toniolo@chu-limoges.fr

**Data Availability Statement:** All relevant data are within the paper and its Supporting Information files.

## Abstract

### Background

31.4 million people in low- and middle-income countries die from chronic diseases annually, particularly in Africa. To address this, strategies such as task-shifting from doctors to nurses have been proposed and have been endorsed by the World Health Organization as a potential solution; however, no comprehensive review exists describing the extent of nurse-led chronic disease management in Africa.

### Aims

This study aimed to provide a thorough description of the current roles of nurses in managing chronic diseases in Africa, identify their levels of knowledge, the challenges, and gaps they encounter in this endeavor.

### Methods

We performed a scoping review following the key points of the Cochrane Handbook, and two researchers independently realized each step. Searches were conducted using five databases: MEDLINE, PyscINFO, CINAHL, Web of Science, and Embase, between October 2021 and April 2023. A descriptive analysis of the included studies was conducted, and the quality of the studies was assessed using the Downs and Black Scale.

### Results

Our scoping review included 111 studies from 20 African countries, with South Africa, Nigeria, and Ghana being the most represented. Findings from the included studies revealed varying levels of knowledge. Nurses were found to be actively involved in managing common chronic diseases from diagnosis to treatment. Facilitating factors included

**Funding:** The authors received no specific funding for this work.

**Competing interests:** The authors have declared that no competing interests exist.

comprehensive training, close supervision by physicians, utilization of decision trees, and mentorship. However, several barriers were identified, such as a shortage of nurses, lack of essential materials, and inadequate initial training.

## Conclusion

There is significant potential for nurses to enhance the screening, diagnosis, and treatment of chronic diseases in Africa. Achieving this requires a combination of rigorous training and effective supervision, supported by robust policies. To address varying levels of knowledge, tailored training programs should be devised. Further research is warranted to establish the effectiveness of nurse-led interventions on population health outcomes.

## Introduction

Africa is currently facing a double challenge linked to epidemiological transition [1, 2]. This transition involves the emergence of chronic diseases in an environment where infectious diseases continue to prevail [2]. Contributing factors include changes in health habits, such as tobacco and alcohol use, as well as economic development and urbanization. As the prevalence of chronic diseases increases, they significantly impact health outcomes and the healthcare system [1, 2].

Chronic diseases, also known as non-communicable diseases (NCDs), are characterized by their non-transmissible occurrence and duration, lasting for three months or more [3]. The World Health Organization (WHO) defines non-communicable diseases as diseases that do not spread from person to person and typically have a long disease duration with slow progression [4]. The main types of NCDs include cardiovascular diseases, cancer, chronic respiratory diseases, and diabetes, with neurological disorders and mental health issues often included in this category. Annually, chronic diseases account for the death of 41 million people worldwide, representing approximately 71% of global deaths, with three-quarters of these deaths (31.4 million) occurring in low- and middle-income countries, notably in Africa [5, 6].

Efficient management of these pathologies with modifiable risk factors necessitates access to care and prevention measures to mitigate their impact on the healthcare system. However, a key challenge faced by African health systems is the shortage of healthcare personnel, with limited resources and an overwhelming workload [7]. For instance, there are only 0.29 doctors and 1.88 nurses per 1000 people in the region [8, 9].

In line with the goal of universal health coverage by 2030, the WHO acknowledges that nurses could be a viable solution to help manage chronic diseases [10–12]. Implementing task-shifting strategies from physicians to nurses is safe and effective in chronic disease management [13, 14]. Task shifting is defined as the transfer of tasks from doctors to nurses [13]. However, the role of nurses in chronic disease management is not limited to this. Chronic disease management activities in primary care include tasks such as taking vital signs, education on healthy diets, lifestyle advice, chronic disease education, and chronic disease clinics [15]. In northern countries, nurses provide NCDs care in a variety of settings, independently and collaboratively, with clinical, managerial, and accountability responsibilities [15, 16]. Furthermore, various strategies have already been tested in the African context to manage infectious diseases, such as task-shifting from doctors to nurses. This approach proved successful in HIV management, where nurses' role enhanced antiretroviral therapy access, and improved patient retention, knowledge, and comfort [13, 17].

Nurses already contribute to managing chronic diseases in Africa, even though it is pertinent to emphasize that there are no systematic country-specific definitions for nursing care in the management of chronic diseases in Africa [18]. Indeed not all countries have policies in place to regulate the nursing profession and nursing care [18]. To define nursing and standardize nursing education and training programs, the West African Health Organization proposed a specific curriculum in 2014 to guide the nursing profession with ten core competencies [19]. These competencies encompass the analysis of functions related to nursing and obstetrical practices; the handling of clinical situations by referring to the physiology and diseases of the human body; intervention in community health; application of methods and techniques of nursing and obstetrics; application of research processes; management of nursing and obstetrical care; promotion, prevention, and rehabilitative care for the "mother and child; administration of nursing care; management of medico-surgical pathologies; and provision of culturally acceptable and high-quality care during pregnancy, childbirth, and delivery, with the ability to manage specific emergency situations to maximize the health of women and their newborns [19]. To note, the program allocated only 20 hours of coursework, specifically for chronic diseases raising questions on nurse knowledge and involvement in chronic disease management. To our knowledge, no comprehensive review has described their roles and knowledge in this context. Consequently, there is a pressing need to clearly outline the role of nurses in chronic disease management and their level of knowledge in Africa. This would facilitate the design of national strategies for nurse-led chronic disease management.

Our review question was: *What were the nurses' roles and knowledge in managing NCDs in Africa*?

We aimed to provide a thorough description of the current roles of nurses in managing NCDs in Africa and identify their levels of knowledge, challenges, and gaps they encounter.

## Methods

We conducted a scoping review [20] following the key points outlined in the Cochrane Handbook [21] from October 01, 2021, to February 10, 2022, with an update conducted before publication, extending until April 30, 2023. This study and its protocol were registered in the PROSPERO database (CRD42021267403). This study was reported following the Preferred Items for Systematic Reviews and Meta-Analyses (PRISMA) guidelines [22].

### Search strategy

To identify relevant articles for review, five databases were searched: MEDLINE, Web of Science, Embase, CINALH, and PyscINFO [23].

The search strategy used a combination of subject headings and keywords of the main topics of the review: nursing, noncommunicable diseases, and Africa. Additionally, a "title and abstract" search was performed for MEDLINE. The search strategy was adapted to suit the specificities of each database for the others. The search strategy used for each database are available in S1 Table.

**Inclusion and exclusion criteria.**   Based on the review question, the inclusion criteria were defined using the PICOS framework. For the "population" item, all studies focusing on nurses' NCDs management, nurses' knowledge of NCDs, or adult patients receiving nursing intervention for NCDs management in Africa were included. Regarding the "intervention" item, studies describing all nurse-led activities regarding chronic diseases [i.e., tasks done by nurses to manage chronic disease, including those that are habitually performed by physicians [13]) or nurses' chronic diseases knowledge evaluation were included. The "comparison" item was not defined because our review did not focus solely on interventional studies. The

"Outcomes" items were defined as the efficacy of nursing interventions, nurses' level of knowledge regarding NCDs, and gaps and challenges linked to nurses' NCDs management. For the "study design" item, we included all study designs relevant to our topic. No language or publication date exclusion criteria were applied. Studies that did not specify specific outcomes for nurses were excluded. Studies were also excluded if they were incomplete study protocols or ongoing studies, editorials, conference proceedings, or if they were published in non-peer-reviewed journals.

## Selection and data collection processes

Rayyan QRCI tool was used to export all identified articles and remove duplicates.

The study selection process was conducted independently by two researchers who were unaware to each other's decisions in the two phases. Initially, studies were identified by reading the titles and abstracts, and eligible articles were then fully read. If disagreements arose, a third researcher was consulted.

## Data extraction and data synthesis

Data extraction was systematically performed using an Excel spreadsheet that included information such as the first author's name, publication year, journal, country where the study was conducted, main objective, study design, disease, intervention/nursing care of interest, sample characteristics, results regarding the intervention/impact of nursing care, facilitators, barriers, and study limitations identified by the authors. A descriptive synthesis was employed, and no meta-analysis was performed. Subgroup analyses were carried out based on disease type and study location.

## Quality assessment

The bias and quality of each study were evaluated. Quantitative studies were evaluated using the Downs and Black scale[24]. Qualitative studies, were evaluated using the specific EQUATOR guidelines [25].

Ethical approval was not required for this study.

## Results

The PRISMA flowchart summarizing the selection process is shown in Fig 1. Following the removal of duplicates, we screened a total of 3,464 articles. Of these, 3,009 were excluded as their titles and abstracts did not align with our specified criteria. Among the remaining 455 articles, a thorough review was conducted, leading to the identification of 111 articles that met our inclusion criteria and were subsequently included into this scoping review.

## Description of the selected studies

S2 Table describes the characteristics of each study. Among the 111 studies, 42 had a cross-sectional design, 19 were qualitative, 12 were prospective interventional non-randomized, 10 were retrospective data analysis, eight were randomized (controlled, cluster, or pilot), eight were quasi-experimental, five were descriptive, four were cohort studies, and three were mixed-method. Publication years ranged from 1985 to 2023. The results of the studies concerned 6,031 nurses and 182,791 patients from 20 African countries. The repartition of the studies in Africa is shown in Fig 2. South Africa was the country with the most important number of studies related to our review (n = 28), followed by Nigeria (n = 16), and Ghana

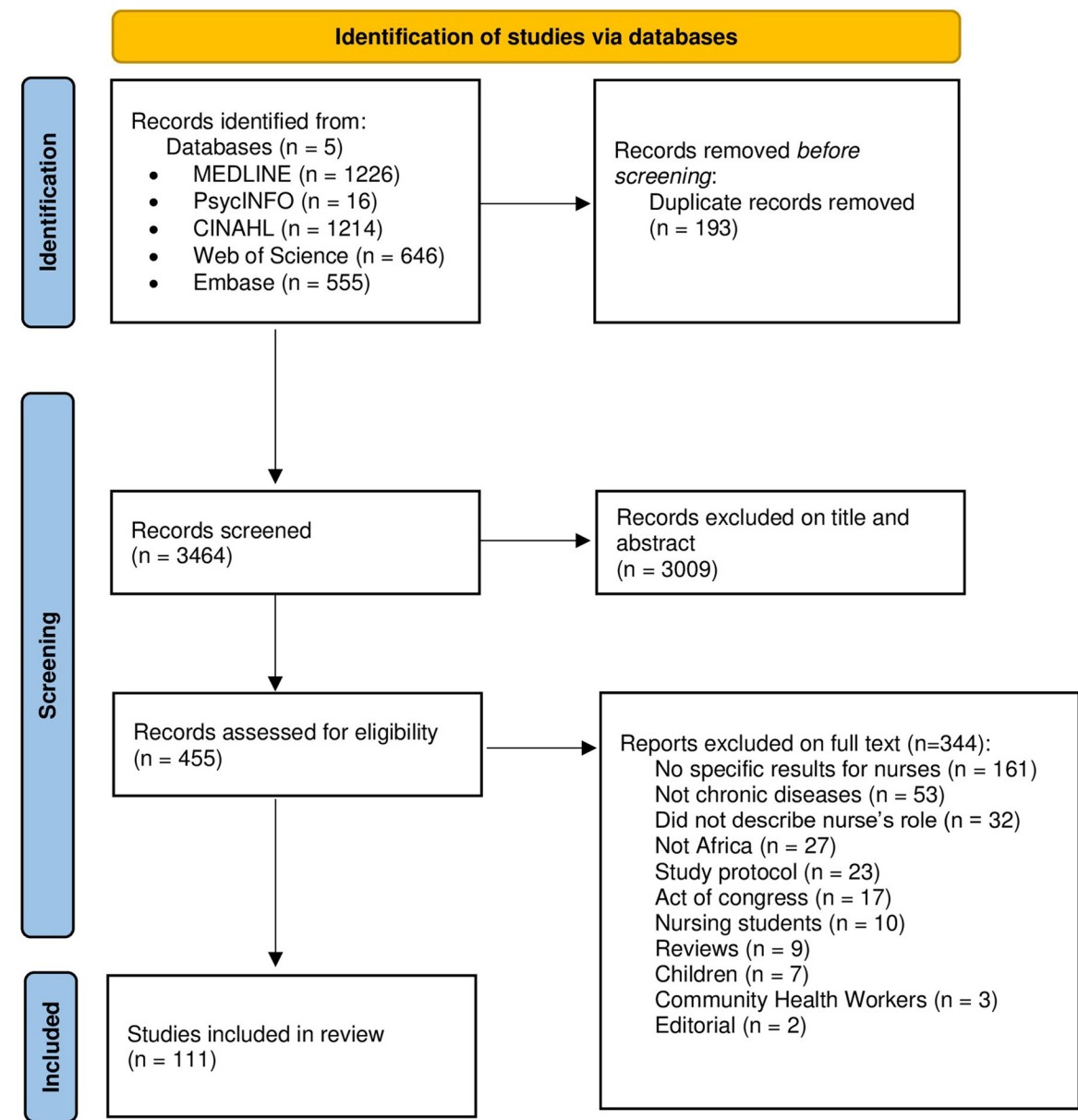

**Fig 1. PRISMA flowchart.** From: http://www.prisma-statement.org/.

(n = 12). The studies included in our review were mainly conducted in English-speaking African countries.

## Quality assessment and limitations of the included studies

Overall, we found low-quality scores among the quantitative included studies. The median score for the Downs and Black scale was 14 (IQR: 11–17). The minimum score was 10, the maximum score was 28. Internal validity was the most concerned subscale for low scores with power calculation. Most of the studies did not calculate the sample size. The detailed Downs and Black scores for each study included in our review are presented in S3 Table. Qualitative

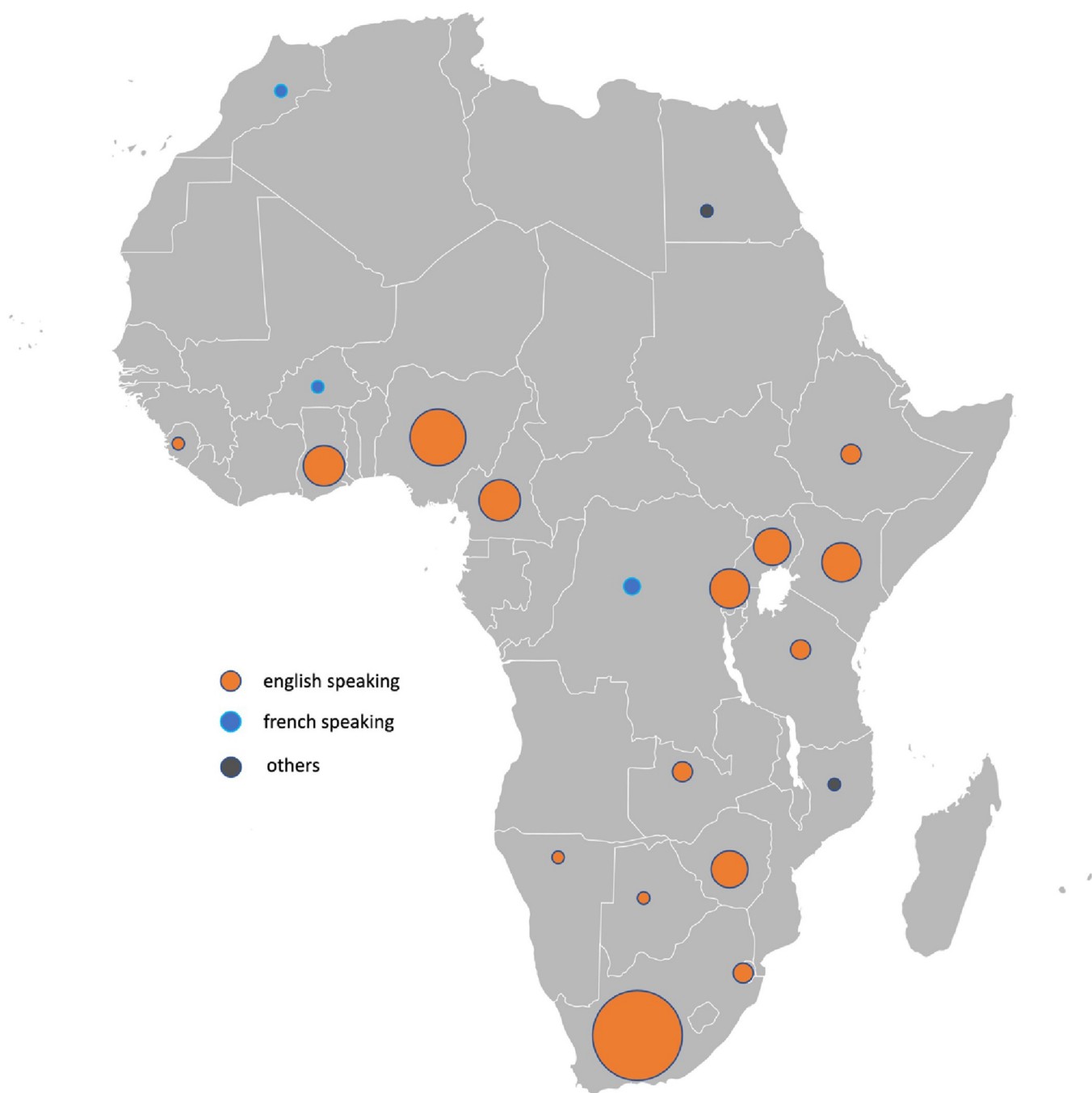

**Fig 2. Distribution of included studies regarding the number of publications and language.** Map credit: https://commons.wikimedia.org/wiki/File:BlankMap-Africa.svg#filelinks.

studies met almost all quality standards of qualitative research regarding appropriate EQUA-TOR guidelines.

S2 Table shows the main limitations of the included studies. They were mainly related to small sample sizes and qualitative, non-randomized, or non-interventional studies.

## Results of the included studies

The studies included in our review were 32.4% focusing on cancer (n = 36), 18% on cardiovascular diseases (n = 20), 17.1% on metabolic diseases (n = 19), 12.6% on mental illness (n = 14), 11.7% on chronic diseases globally (n = 13), 6.3% focusing on neurological diseases (n = 7), and 0.9% on chronic respiratory disease and chronic kidney diseases. S2 Table describes the results of each study.

**Role and knowledge of nursing among chronic diseases.** Among the 36 studies which focused on cancer, 12 were on breast cancer [26–36], 22 on cervical cancer [37–57], one on cancer generally [58] and one on oral cancer [59]. Studies on cervical cancer were focused on nurse-led screening and treatment of human papillomavirus. Screening strategies using nurses were effective, with good agreement with medical doctors. Nurses' level of knowledge varied geographically and in a timely manner among cervical and breast cancer patients, ranging from low- to high-level knowledge. Nurse-led breast cancer screening was also implemented in a lower proportion than cervical cancer screening; however, after training, good examination skills were also identified [34], and a study focused on echography led by nurses with promising results [31].

Cardiovascular diseases were the second major topic of the included studies. There were 17 on high blood pressure [60–75], two on cardiovascular diseases globally [76, 77], and one on heart failure [78]. Nurses' knowledge of high blood pressure varied importantly, ranging from poor to good knowledge after specific training. Nurses obtained a statistically significant improvement in blood pressure; they independently screened and managed blood pressure after training, following decision trees in rural and urban areas. Echocardiography was used to screen for heart failure in a study on nurse-led heart failure screening [78].

Diabetes exclusively represented metabolic diseases [79–97]. Nurses had varying levels of knowledge about diabetes and its complications ranging from poor associated with the misconception that diabetes was a contagious disease as an example [80] to good knowledge associated with improved diabetes management [83, 84, 86, 91]. In most of the studies, nurses were trained in diagnosis and monitoring including prescribing treatments and augmentation for diabetes. Decision trees and algorithms were used for this purpose.

Thirteen studies focused on managing or knowing chronic diseases globally, or among multiple chronic diseases [85, 98–109]. Nurses' roles were similar to those described above: they managed chronic diseases, followed training with algorithms or decision trees, and supervision. They obtained positive outcomes for high blood pressure, diabetes, asthma, diabetes, sickle cell, and adherence [98, 99, 102, 104, 105, 108, 109].

Regarding mental illness, seven studies focused on mental illness globally [110–116], six on depression [117–122], and one on schizophrenia [123]. Globally, there was a gap in nurses' mental health knowledge. They improved their knowledge with training and were able to perform psychological interventions, screening, and initiation of treatment improving the mental health status and satisfaction of patients.

Studies on neurological diseases included epilepsy (n = 4) [124–127], and post-stroke dysphagia (n = 3) [128–130]. Studies on post-stroke dysphagia were focused on nurses' attitudes and knowledge. They found a low level of knowledge regarding the topic including diagnosis, evaluation, and treatment. Regarding epilepsy, these were interventional studies, where nurses could, after training, effectively manage epilepsy (diagnosis, treatment, and follow-up).

The two topics that were least covered were respiratory and chronic kidney diseases. A study focused on nurse-led management of asthma [131] was associated with an improvement in asthma condition in patients. One study focused on nurses' level of knowledge of chronic kidney disease which remained moderate [132].

**Challenges and gaps to nurse-led chronic diseases management.** S2 Table shows the details of the challenges and gaps identified in our review. To summarize, the main challenges identified in our review to nurse-led chronic disease management were linked to the initial nurses' knowledge gap regarding chronic diseases. The different curricula of the nurses did not include sufficient basic knowledge in each of our categories of chronic diseases. Other barriers were linked to the cost of care, shortage of nurses, lack of materials, and lack of transportation for patients.

Facilitators were also identified as key components of a well-done nurses' task-shifting strategy. The first important facilitator was training; moreover, regular training, supervision by physicians, and mentorship were the key points associated with the effective implementation of nurse-led strategies. The training should include both theoretical and practical sessions, in which training varied widely between studies, from one session for half a day to multiple and repeated sessions. Moreover, the use of decision trees or algorithms was associated with positive patient outcomes. Using an already existent system, such as HIV clinics was also a favorable strategy to implement chronic diseases care.

## Discussion

To the best of our knowledge, this scoping review represents a pioneering effort conducted in Africa to comprehensively delineate the roles and knowledge of nurses in managing NCDs. The findings of our review indicate that nurses possess varying levels of knowledge; however post-training, they demonstrate the capability to enhance the detection, treatment, and overall management of most chronic diseases. Effective task-shifting programs are achieved through essential components such as comprehensive training, ongoing supervision, and the use of decision trees, ensuring the quality of nurse-led interventions. While our review covered the major categories of chronic diseases, it is crucial to note that several subjects, such as Chronic Obstructive Pulmonary Disease and Alzheimer's disease, received limited or no investigation in Africa.

The varying levels of knowledge among nurses about different chronic diseases, ranging from low to high, reflect the gaps in their initial training and curricula. This was also found in international literature for diabetes [133] as examples. Not all nursing diplomas in Africa are at the Bachelor's level. Harmonizing the quality of higher education in nursing, as initiated by the African Union [134], is essential to improve the quality of care [135]. Increasing the level of training to the Bachelor's level and incorporating chronic disease management into the curriculum can be instrumental in enhancing nurses' capacity to manage NCDs effectively [10, 134].

Despite the variability in knowledge, our review demonstrated improved patient outcomes resulting from nursing interventions for chronic diseases. This observation aligns with existing evidence from task-shifting strategies implemented in the context of infectious diseases in Africa [17]. Numerous studies have already shown the effectiveness of nurses' task-shifting strategies [14], such as managing cardiovascular diseases [136], detecting cancer [137], and providing mental health support [138, 139]. These strategies are encouraged by the WHO to enhance NCDs management [7, 10].

The challenges and gaps identified in our review were consistent with the literature on implementation science. Adequate training, well-designed strategies, supervision, funding, leadership, and sufficient resources are the key facilitators of successful implementation [140, 141]. Addressing these factors can lead to improved chronic disease management outcomes in the African healthcare systems.

The different task-shifting strategies identified in our review have prompted us to consider the potential of expanding nurses' roles in Africa. The roles described in the studies closely

align with those of advanced practice nursing notably regarding their clinical skills [142–144]. Implementing advanced practice nursing in addition to training nurses in task-shifting strategies could be an effective or complementary solution. Although the implementation of advanced practice nursing has begun in some African countries such as South Africa and Kenya, it remains limited [145]. Scaling up the implementation of advanced practice nursing across African countries could significantly improve chronic disease management [146] and population outcomes [143].

This study has important implications for public health. First, we emphasize the potential of task shifting to nurses to enhance healthcare access, particularly in regions with healthcare professional shortages. We also highlight the effectiveness of nurse-led chronic disease management, which supports the argument for efficient healthcare models, particularly in resource-constrained settings, where cost-effectiveness is paramount. Our research also indicates that nurse-led care can have a positive impact on patient satisfaction and outcomes, advocating for a shift towards patient-centered care models. This transformation is critical for improving management of chronic diseases. Nurse-led care also enhances continuity, ensuring that patients receive sustained support, education, and follow-up. This is of utmost importance in the effective management of chronic conditions and prevention of complications. Furthermore, our study highlighted the multifaceted role of nurses, extending beyond individual patient care to community health promotion. Nurses' involvement in addressing the risk factors for chronic diseases at the community level is crucial. Moreover, our research suggests potential avenues for the development of training programs and educational initiatives designed to equip nurses with the skills and knowledge necessary for effective chronic disease management. Furthermore, our study calls for policymakers in each country to formally define, recognize, and support nurses' roles in chronic disease management. This recognition could lead to the establishment of improved regulatory frameworks and increased resources for nurse-led care. Finally, our research contributes to data-driven decision-making within healthcare systems, enabling administrators to make informed choices regarding resource allocation.

## Limitations

The results of our review must be interpreted with caution because of the relatively low quality of the included studies, particularly concerning their external validity and heterogeneity. Despite encountering studies with low scores in the Downs and Black assessment, we deliberately chose to include all the identified studies in our analysis. This decision stems from the nature of scoping reviews as a form of evidence synthesis that seeks to systematically identify and map the breadth of available evidence on a specific topic. Unlike other review types, the rigorous assessment of methodological quality is not an absolute prerequisite for scoping reviews. However, in our commitment to enhancing the quality and credibility of our research, we undertook the assessment to evaluate the methodological robustness of the included studies. This assessment was carried out to elevate the overall rigor of our study and to provide readers with a discerning evaluation of the studies' evidentiary foundation. By undertaking this assessment, our intention was to ensure a high level of transparency, integrity, and meticulousness throughout our review process. More robust studies with randomized controlled designs are needed to ascertain the effectiveness of nurse-led interventions compared to usual care. Additionally, most of the included studies were conducted in English-speaking African countries, possibly influenced by the historical link between these countries and North America, where nurses have had an expanded role and have developed university nursing curricula and research partnerships [147].

## Conclusions

We provide a comprehensive overview of nurses' knowledge and role in managing NCDs using a robust scoping review methodology, making it the first in Africa. Our findings serve as a guide for future research and offer recommendations for developing nursing programs in Africa.

Nurses can play a crucial role in addressing the shortage of doctors and the treatment gap in managing chronic diseases. However, nurses require robust training, ongoing supervision, supportive environments, and strong policies to fulfill this potential. Further research with a robust study design is necessary to conclusively assess the effectiveness of these strategies. African countries employing task-shifting strategies should consider the implementation of advanced practice nursing, as evidenced by successful examples such as the management of diabetes in South Africa.

## Supporting information

**S1 Table. Search strategy for each database included in the review.**
(DOCX)

**S2 Table. Description of the included studies.** Studies were classified according to disease type and in chronological order.
(DOCX)

**S3 Table. Downs and Black assessment for included studies.**
(DOCX)

**S1 Checklist. Preferred Reporting Items for Systematic reviews and Meta-Analyses extension for Scoping Reviews (PRISMA-ScR) checklist.**
(DOCX)

## Author Contributions

**Conceptualization:** Jean Toniolo, Edgard Brice Ngoungou, Pierre-Marie Preux, Pascale Beloni.

**Data curation:** Jean Toniolo, Edgard Brice Ngoungou, Pascale Beloni.

**Formal analysis:** Jean Toniolo, Edgard Brice Ngoungou.

**Investigation:** Jean Toniolo, Edgard Brice Ngoungou, Pierre-Marie Preux, Pascale Beloni.

**Methodology:** Jean Toniolo, Edgard Brice Ngoungou, Pierre-Marie Preux, Pascale Beloni.

**Software:** Jean Toniolo.

**Supervision:** Edgard Brice Ngoungou, Pierre-Marie Preux, Pascale Beloni.

**Validation:** Jean Toniolo, Edgard Brice Ngoungou, Pierre-Marie Preux, Pascale Beloni.

**Visualization:** Jean Toniolo, Edgard Brice Ngoungou, Pierre-Marie Preux, Pascale Beloni.

**Writing – original draft:** Jean Toniolo, Edgard Brice Ngoungou, Pierre-Marie Preux, Pascale Beloni.

**Writing – review & editing:** Jean Toniolo, Edgard Brice Ngoungou, Pierre-Marie Preux, Pascale Beloni.

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
