## [Decision Letter · Decision Letter 0]

11 Jul 2023

PONE-D-23-15682Role and knowledge of nurses in the management of non-communicable diseases in Africa: a scoping reviewPLOS ONE

Dear Dr. Toniolo,

Thank you for submitting your manuscript to PLOS ONE. After careful consideration, we feel that it has merit but does not fully meet PLOS ONE’s publication criteria as it currently stands. Therefore, we invite you to submit a revised version of the manuscript that addresses the points raised during the review process.

The reviewers have provided feedback on the manuscript to enhance its quality. The authors are advised to consider the following four key points for any subsequent revisions:

Conduct thorough English proofreading to ensure that the manuscript adheres to proper grammar, style, and language usage.Improve the clarity and alignment of the abstract with the objectives of the study.Enhance the introduction by addressing any inconsistencies and providing a clearer explanation of the role of nurses in the context of the study.Reconsider the inclusion of studies with low quality in your review. Evaluate their impact on the overall credibility and reliability of the findings. If included, we encourage you to provide a rationale for their inclusion or consider excluding them from the study.

In addition to these key points, please consider the following suggestions:

Please ensure to reference each supplementary file appropriately within the manuscript.To avoid redundancy, it would be advisable to keep the search terms in the supplementary file.Ensure that the information provided in Supplementary Table 2 is complete.Please verify the accuracy and completeness of your reference list, including all relevant references, such as Higgins et al., 2019, and ensure consistent citation format throughout the manuscript as per the guidelines provided by the journal.We kindly request you to remove any unnecessary components of the main manuscript file, such as the Funding Statement, Declaration of Interests, Authorship Contribution, and Twitter handles. These can be included in the online submission system.

We look forward to receiving your revised manuscript.

Kind regards,

Delfina Fernandes Hlashwayo, M.Sc.

Academic Editor

PLOS ONE

4. We note that Figure 2 in your submission contain [map/satellite] images which may be copyrighted. All PLOS content is published under the Creative Commons Attribution License (CC BY 4.0), which means that the manuscript, images, and Supporting Information files will be freely available online, and any third party is permitted to access, download, copy, distribute, and use these materials in any way, even commercially, with proper attribution. For these reasons, we cannot publish previously copyrighted maps or satellite images created using proprietary data, such as Google software (Google Maps, Street View, and Earth). For more information, see our copyright guidelines: http://journals.plos.org/plosone/s/licenses-and-copyright.

Reviewers' comments:

Reviewer's Responses to Questions

**Comments to the Author**

1. Is the manuscript technically sound, and do the data support the conclusions?

Reviewer #1: Yes

Reviewer #2: No

2. Has the statistical analysis been performed appropriately and rigorously? 

Reviewer #1: Yes

Reviewer #2: N/A

3. Have the authors made all data underlying the findings in their manuscript fully available?

Reviewer #1: Yes

Reviewer #2: Yes

4. Is the manuscript presented in an intelligible fashion and written in standard English?

Reviewer #1: Yes

Reviewer #2: Yes

5. Review Comments to the Author

Reviewer #1: The authors clearly were careful in choosing this topic. The paper definitely provides a solution to a widely asked question on the role of nurses and other health care providers apart from physicians / medical doctors and medical specialists in the diagnosis and management of non-communicable diseases in Africa where they are fast increasing in prevalence due to various life changing landscapes.

However, the authors need to correctly write some of their sentences for them to have a clear meaning to the readers.

Also, the conclusion of the study which clearly stated what role can the nurses assume in these circumstances need to be supported by other strong findings which were not sated in this particular paper. Such issues as what training and for how long needs to be researched and explained in detail for researchers to come to their stated conclusion.

Reviewer #2: I appreciate the opportunity to review the article Role and Knowledge of Nurses in the Management of non-communicable diseases in Africa: a scoping review.

This, without a doubt, is of interest for the development of nursing in the world, as well as it shows us a broad panorama of Nursing in Africa.

However, this document needs to be addressed in many aspects. I believe that if the researchers carry out an in-depth review of the document, as well as pay attention to all the comments, it can improve and be publishable in such a prestigious journal.

I comment on general points in case the researchers wish to attend to them and revise them.

In relation to the summary: specifically the method, this is more than explaining what the researchers did, it must explain the design and search strategy, as well as the sources of information.

The conclusion of the summary does not correspond to the stated objectives, it does not describe the levels of knowledge they have and how they manage non-communicable diseases.

They show nurses as a solution to the lack of doctors, but not to the lack of nurses itself.

Background: page 6 line 125. Place an acronym, after the words: non-communicable disease (NCD).

Page 6 lines 139-143: it mentions twice that task-shifting is used, the paragraph is repeated and it does not mention what they are, only that it worked for HIV management.

At the same time, it does not justify that the lack of doctors is responded to by nurses taking on more activities if there is also a lack of personnel. They are not compensated and there is no mention of interdisciplinary work.

Method: page 7, lines 158 to 163. It shows two types of appointments, it must be unified or in any case, for example. “According to Higgins(quote) in case you want to show the author's name.

Page 7 lines 168 -169. Much use is made of the first-person text, the passive voice is recommended in scientific articles.

On the other hand, in this same section, it is observed that it is improper to show the degrees of the researchers or what they did when this is shown in the contributions of the researcher.

Line 171-173. They could only mention the main topics of the review and mention the attached table in which it is specifically elaborated.

The search equations are removed from the method section and left as supplementary material.

Page 10 method section.: The inclusion and exclusion criteria could be reduced to population and intervention, including in the last part of the objectives that were sought, such as level of knowledge.

It does not include the years that are taken into account

Page 11. Line 251 is peer review, not double blinding

On pages 12 and 13 lines 295-299, I don't know if these results could make a table and point it out or if this comes in table 2, mention it.

Page 13, line 303, You could mention the chronic diseases found or not communicable and later describe them.

Page 15, Did the training last some time, was it given to all the nurses or specifically to a single group?

Also, was the level of knowledge prior to the training assessed in any way? or how to prove that it worked?

Page 16, line 394: use passive voice.

Page 18 line 435 is missing an S.

Page 19 lines 453 to 459: Perhaps advanced practiced nurses because of the knowledge they have, but non-specialized nurses could not be a solution.

I recognize the supplementary material 2 summary table of the 111 articles is too extensive and has something that worries me a little. The document speaks in general about the knowledge that Nursing has, but in these articles, the population is doctors, nurses, and patients. Wouldn't it be exclusive to have different types of targeted populations? It seems that there was no systematization, but if this point is clarified, we will gladly continue.

In supplemental material number 3. I consider that there are very low scores. Because those articles were not removed as they have poor methodological quality.

6. PLOS authors have the option to publish the peer review history of their article (what does this mean?). If published, this will include your full peer review and any attached files.

Reviewer #1: No

Reviewer #2: No

---

## [Author Response · Author response to Decision Letter 0]

23 Aug 2023

Dear Delfina Fernandes Hlashwayo, 

We would like to express our gratitude to you and the esteemed reviewers for dedicating valuable time and providing insightful feedback on our manuscript PONE-D-23-15682 entitled “Role and knowledge of nurses in the management of non-communicable diseases in Africa: a scoping review”. 

Your constructive comments have undoubtedly enhanced the quality and rigor of our research, and we are honored by the opportunity to address these concerns and suggestions.

In this response letter, we have diligently addressed below each reviewer's comments, providing detailed explanations of the revisions we have made to the manuscript. The changes were highlighted in the manuscript. 

Editor comments : 

1. Conduct thorough English proofreading to ensure that the manuscript adheres to proper grammar, style, and language usage.

We performed a complete proofreading of the paper.

2. Improve the clarity and alignment of the abstract with the objectives of the study.

The abstract was modified in alignment with the objectives of the study: p3-4 L60-86.

3. Enhance the introduction by addressing any inconsistencies and providing a clearer explanation of the role of nurses in the context of the study.

We rewrite and completed the introduction : p4-6.

4. Reconsider the inclusion of studies with low quality in your review. Evaluate their impact on the overall credibility and reliability of the findings. If included, we encourage you to provide a rationale for their inclusion or consider excluding them from the study.

We added a paragraph in the Method section (L193-195) and the Limitations section (L359-369). It is important to note that the low-quality scores obtained by qualitative studies were due to the use of the Downs and Black scale, which was initially constructed for quantitative studies, this scale was employed as part of our concerted endeavor to standardize our evaluative approach. To enhance clarity, we removed the assessment of qualitive studies by the Downs and Black scale. We ensured the quality of qualitative studies using appropriate EQUATOR guidelines.

Furthermore, it is essential to clarify the nature of scoping reviews. Scoping reviews are a type of evidence synthesis that aims “to systematically identify and map the breadth of evidence available on a particular topic, field, concept, or issue, often irrespective of the source (i.e., primary research, reviews, non-empirical evidence) within or across specific contexts. Scoping reviews can clarify key concepts and definitions in the literature and identify key characteristics or factors related to a concept, including those related to methodological research”(Munn, 2022, JBI evidence synthesis). Moreover, regarding the assessment of the methodological quality of included studies, it is not a strict requirement for scoping reviews. However, we performed this assessment to elevate the quality of our research work and to provide a critical appraisal for readers regarding the level of evidence of the studies' findings. By conducting this assessment, we aimed to ensure transparency and rigor in our review process. Taken together, these arguments make it possible to maintain all the studies included in our review.

In addition to these key points, please consider the following suggestions:

1. Please ensure to reference each supplementary file appropriately within the manuscript.

It was done.

2. To avoid redundancy, it would be advisable to keep the search terms in the supplementary file.

It was done.

3. Ensure that the information provided in Supplementary Table 2 is complete.

The information provided in supplementary table 2 is complete. 

4. Please verify the accuracy and completeness of your reference list, including all relevant references, such as Higgins et al., 2019, and ensure consistent citation format throughout the manuscript as per the guidelines provided by the journal.

We adjusted the reference list and verified citation format throughout the manuscript.

5. We kindly request you to remove any unnecessary components of the main manuscript file, such as the Funding Statement, Declaration of Interests, Authorship Contribution, and Twitter handles. These can be included in the online submission system.

It was done. 

1. Please ensure that your manuscript meets PLOS ONE's style requirements

We modified the manuscript. 

2. In your Data Availability statement, you have not specified where the minimal data set underlying the results described in your manuscript can be found. PLOS defines a study's minimal data set as the underlying data used to reach the conclusions drawn in the manuscript and any additional data required to replicate the reported study findings in their entirety. 

S1 and S2 tables are available with the manuscript, they provide essential information that allows for the replication of the study findings.

It was done. 

4. We note that Figure 2 in your submission contain [map/satellite] images which may be copyrighted. 

We have modified Fig 2 using a non-copyrighted map from the public domain source: https://commons.wikimedia.org/wiki/File:BlankMap-Africa.svg#filelinks.

5. Please include captions for your Supporting Information files at the end of your manuscript, and update any in-text citations to match accordingly. 

It was done: p26

Reviewer #1:

“The authors clearly were careful in choosing this topic. The paper definitely provides a solution to a widely asked question on the role of nurses and other health care providers apart from physicians / medical doctors and medical specialists in the diagnosis and management of non-communicable diseases in Africa where they are fast increasing in prevalence due to various life changing landscapes.”

Thank you for your positive comments.

“However, the authors need to correctly write some of their sentences for them to have a clear meaning to the readers.”

We performed a complete proofreading of the paper.

“Also, the conclusion of the study which clearly stated what role can the nurses assume in these circumstances need to be supported by other strong findings which were not sated in this particular paper. Such issues as what training and for how long needs to be researched and explained in detail for researchers to come to their stated conclusion.”

We modified the conclusion section (P16 L374-385) and precise in the results sections for training (P13 L297-298).

Reviewer #2:

“I appreciate the opportunity to review the article Role and Knowledge of Nurses in the Management of non-communicable diseases in Africa: a scoping review.

This, without a doubt, is of interest for the development of nursing in the world, as well as it shows us a broad panorama of Nursing in Africa.

However, this document needs to be addressed in many aspects. I believe that if the researchers carry out an in-depth review of the document, as well as pay attention to all the comments, it can improve and be publishable in such a prestigious journal.

I comment on general points in case the researchers wish to attend to them and revise them.”

Thank you for your positive comments.

“In relation to the summary: specifically the method, this is more than explaining what the researchers did, it must explain the design and search strategy, as well as the sources of information.

The conclusion of the summary does not correspond to the stated objectives, it does not describe the levels of knowledge they have and how they manage non-communicable diseases.

They show nurses as a solution to the lack of doctors, but not to the lack of nurses itself.”

We modified the abstract according to these suggestions: P3-4 L60-86.

“Background: page 6 line 125. Place an acronym, after the words: non-communicable disease (NCD).”

It was done and the acronym was used throughout the manuscript.

“Page 6 lines 139-143: it mentions twice that task-shifting is used, the paragraph is repeated and it does not mention what they are, only that it worked for HIV management.

At the same time, it does not justify that the lack of doctors is responded to by nurses taking on more activities if there is also a lack of personnel. They are not compensated and there is no mention of interdisciplinary work.”

We modified this paragraph, we clearly describe the role of nurses for HIV and explained the role of nurses for chronic disease management as well as the fact that they work independently or collaboratively. (P5-6 L119-128)

“Method: page 7, lines 158 to 163. It shows two types of appointments, it must be unified or in any case, for example. “According to Higgins(quote) in case you want to show the author's name.”

It was modified.

“Page 7 lines 168 -169. Much use is made of the first-person text, the passive voice is recommended in scientific articles.”

We modified the method section using passive voice. (P6-9)

“On the other hand, in this same section, it is observed that it is improper to show the degrees of the researchers or what they did when this is shown in the contributions of the researcher.”

We withdrew the sentence. 

“Line 171-173. They could only mention the main topics of the review and mention the attached table in which it is specifically elaborated.

The search equations are removed from the method section and left as supplementary material.”

It was done. 

“Page 10 method section.: The inclusion and exclusion criteria could be reduced to population and intervention, including in the last part of the objectives that were sought, such as level of knowledge.”

We have modified this paragraph, but still using the PICOS method, which provides greater clarity from our point of view. (P7-8 L 161-175).

“It does not include the years that are taken into account.”

It was noted that “No language or publication date exclusion criteria were applied” (P8 L 173).

“Page 11. Line 251 is peer review, not double blinding”

Each investigator was unaware to the other's decisions regarding the inclusion of articles and throughout the other steps of the literature review, We modified the sentence. (P8 L179-180). 

“On pages 12 and 13 lines 295-299, I don't know if these results could make a table and point it out or if this comes in table 2, mention it.”

Results are available in S2 Table, we mentioned it: P10 L227.

“Page 13, line 303, You could mention the chronic diseases found or not communicable and later describe them.”

We modified the paragraph but as the study focused on the different diseases we prefer immediately mentioned them. (P10 L231)

“Page 15, Did the training last some time, was it given to all the nurses or specifically to a single group? Also, was the level of knowledge prior to the training assessed in any way? or how to prove that it worked?”

It is difficult to summarize the content of all the training programs for all the studies in a few lines, so we have modified the paragraph and added some clarifications, as for the other studies, the study design and content of the training programs are available in the S2 Table. (P13 L 293 – 301)

“Page 16, line 394: use passive voice.”

It was done. (P14 L327)

“Page 18 line 435 is missing an S.”

It was modified. 

“Page 19 lines 453 to 459: Perhaps advanced practiced nurses because of the knowledge they have, but non-specialized nurses could not be a solution.”

I agree with you but advanced practice nursing is not the only solution to address the healthcare challenges. Registered nurses, with appropriate training and support, can indeed play a crucial role in task-shifting strategies. By expanding the scope of practice for registered nurses and providing them with additional training in specific areas, they can effectively take on tasks traditionally performed by advanced practice nurses or physicians. Task-shifting to nurses is a viable and effective approach in various healthcare settings, especially in regions with limited healthcare resources, such as many parts of Africa. 

We modified the conclusion. (P16 L374-386)

“I recognize the supplementary material 2 summary table of the 111 articles is too extensive and has something that worries me a little. The document speaks in general about the knowledge that Nursing has, but in these articles, the population is doctors, nurses, and patients. Wouldn't it be exclusive to have different types of targeted populations? It seems that there was no systematization, but if this point is clarified, we will gladly continue.”

We specified this point in the method (P7-8 L172-176). As described in the method, we included all studies that specifically examined nurses' involvement in chronic disease management or assessed nurses' knowledge related to chronic diseases. We excluded studies that did not provide explicit outcomes related to nurses to maintain a focused and relevant analysis. However, it is important to note that the studies we considered could involve not only nurses but also patients or doctors. As part of our comprehensive approach, we have reported the complete composition of the study population.

“In supplemental material number 3. I consider that there are very low scores. Because those articles were not removed as they have poor methodological quality.”

As explained above, very low scores were linked to qualitative design. We added a paragraph in the Method section (L193-195) and the Limitations section (L359-369). It is important to note that the low-quality scores obtained by qualitative studies were due to the use of the Downs and Black scale, which was initially constructed for quantitative studies. Despite receiving low scores, these qualitative studies were of good quality and adhered to the standards for qualitative research. This scale was employed as part of our concerted endeavor to standardize our evaluative approach. To enhance clarity, we removed the assessment of qualitive studies by the Downs and Black scale. We ensured the quality of qualitative studies using appropriate EQUATOR guidelines.

We believe that the revisions we have implemented significantly improve the overall clarity and impact of our work. The revised manuscript now aligns more closely with the high standards set by PLOS One, and we are confident that it will contribute meaningfully to the scientific community.

Once again, we express our gratitude for the guidance provided by the reviewers and yourself. We are committed to ensuring the highest quality of our research and welcome any further suggestions for improvement.

Please find attached the revised version of the manuscript.

Thank you for your continued support, and we eagerly await your decision on our revised submission.

Sincerely

---

## [Decision Letter · Decision Letter 1]

10 Oct 2023

PONE-D-23-15682R1Role and knowledge of nurses in the management of non-communicable diseases in Africa: a scoping reviewPLOS ONE

Dear Dr. Toniolo,

Thank you for submitting your manuscript to PLOS ONE. After careful consideration, we feel that it has merit but does not fully meet PLOS ONE’s publication criteria as it currently stands. Therefore, we invite you to submit a revised version of the manuscript that addresses the points raised during the review process.

In addition to the comments made by the two reviewers, please address the following points in the revised version of the manuscript:

**General comments:**

Please review the writing in the entire manuscript to ensure there are no grammatical errors, typos, or any other issues.Please review the abbreviation: NCDs stand for non-communicable diseases, while 'NCD' is used for the singular term.

**Methods:**

Line 146: Consider capitalizing "PROSPERO" for consistency.Line 157: In the context of systematic reviews and database searches, the term "equation" is not commonly used. Instead, the more appropriate term is "search strategy."Lines 204-205: Please review the sentence construction for improved clarity.

**Results:**

Line 225: Replace "6,3%" with "6.3%" for standard numeric representation.Line 304: Consider using "overall" instead of "globally" for better precision.Reevaluate the placement of the quality assessment in the results section; it might be more suitable to position it elsewhere rather than at the end.

**Discussion:** Please expand on the implications of your results for public health.

**Conclusions:**

Consider beginning without the phrase 'despite these limitations'.Lines 383-384: Instead of "will guide," consider "serves as a guide for future research."It is not customary to include citations in the conclusions section.

**Figures and Supplementary files:**

Fig 2: Please rephrase the caption for better clarity. Consider using a term other than "repartition" for improved adequacy.S2 Table: Consider repeating header rows for enhanced clarity.

We look forward to receiving your revised manuscript.

Kind regards,

Delfina Fernandes Hlashwayo, M.Sc.

Academic Editor

PLOS ONE

Journal Requirements:

Reviewers' comments:

Reviewer's Responses to Questions

**Comments to the Author**

1. If the authors have adequately addressed your comments raised in a previous round of review and you feel that this manuscript is now acceptable for publication, you may indicate that here to bypass the “Comments to the Author” section, enter your conflict of interest statement in the “Confidential to Editor” section, and submit your "Accept" recommendation.

Reviewer #3: All comments have been addressed

Reviewer #4: All comments have been addressed

2. Is the manuscript technically sound, and do the data support the conclusions?

Reviewer #3: Partly

Reviewer #4: Yes

3. Has the statistical analysis been performed appropriately and rigorously? 

Reviewer #3: N/A

Reviewer #4: Yes

4. Have the authors made all data underlying the findings in their manuscript fully available?

Reviewer #3: No

Reviewer #4: Yes

5. Is the manuscript presented in an intelligible fashion and written in standard English?

Reviewer #3: Yes

Reviewer #4: Yes

6. Review Comments to the Author

Reviewer #3: This manuscript provides a comprehensive overview to help guide the management of NCDs in Africa. There are a few recommendations. In the introduction section, the authors should articulate the nursing roles for the management of NCDs as defined by nursing organizations within the country. This will enable a thorough discussion of this information with the study's findings. Additionally, it would be advantageous to provide a clear description of the levels of evidence present in the included studies.

Reviewer #4: Reviwer’s comments

The authors have chosen a very interesting topic. The world is faced with a growing number of chronic diseases.

• In line 123, the sentence seems incomplete , I suggest that the authors complete the sentence

• In line 135 – well done to the authors -the review question is relevant in achieving the aim and objectives of the study.

• In terms of the results section, I suggest that the authors should start by describing the types of the studies reviewed, then present the reviewed results . the contents in line 210-219 may have been presented before the roles and knowledge of the nurses.

• It should have been of importance if the findings from the review were presented in themes , the reader of the manuscripts should know the types of roles and the knowledge found in the review.

• In line 282-300, the authors presented the barriers and facilitators ,where as in the aims of the study the authors talked about challenges and gaps. This creates a confusion. It feels like it is a two in one study. I suggest the authors be consistent with the words used.

• Line 315-324, the paragraph does not have reference, the starting of the paragraph starts with “To our knowledge …” which may show assumption , I believe there has been more reviews , this review may be adding to the available or bringing in new knowledge. I suggest that the authors look into paragraph.

• The study reviewed 111 studies, however in line 343-247,only two studies were included in relation to the barriers and facilitators.

7. PLOS authors have the option to publish the peer review history of their article (what does this mean?). If published, this will include your full peer review and any attached files.

Reviewer #3: No

Reviewer #4: No

---

## [Author Response · Author response to Decision Letter 1]

16 Oct 2023

Dear Delfina Fernandes Hlashwayo, 

We would like to express our gratitude to you and the esteemed reviewers for dedicating valuable time and providing insightful feedback on our manuscript PONE-D-23-15682R1 entitled “Role and knowledge of nurses in the management of non-communicable diseases in Africa: a scoping review”. 

Your constructive comments have undoubtedly enhanced the quality and rigor of our research, and we are honored by the opportunity to address these concerns and suggestions.

In this response letter, we have diligently addressed below each reviewer's comments, providing detailed explanations of the revisions we have made to the manuscript. The changes were highlighted in the manuscript. 

Editor comments : 

General comments:

1. Please review the writing in the entire manuscript to ensure there are no grammatical errors, typos, or any other issues.

It was done.

2. Please review the abbreviation: NCDs stand for non-communicable diseases, while 'NCD' is used for the singular term.

It was done.

Methods:

1. Line 146: Consider capitalizing "PROSPERO" for consistency.

It was done.

2. Line 157: In the context of systematic reviews and database searches, the term "equation" is not commonly used. Instead, the more appropriate term is "search strategy."

We have modified it.

3. Lines 204-205: Please review the sentence construction for improved clarity.

We modified this sentence: “Following the removal of duplicates, we screened a total of 3,464 articles. Of these, 3,009 were excluded as their titles and abstracts did not align with our specified criteria. Among the remaining 455 articles, a thorough review was conducted, leading to the identification of 111 articles that met our inclusion criteria and were subsequently included into this scoping review.”

Results:

1. Line 225: Replace "6,3%" with "6.3%" for standard numeric representation.

It was done.

2. Line 304: Consider using "overall" instead of "globally" for better precision.

We modified it.

3. Reevaluate the placement of the quality assessment in the results section; it might be more suitable to position it elsewhere rather than at the end.

We modified the order of the result section.

4. Discussion: Please expand on the implications of your results for public health.

We added a paragraph at the end of the discussion: “This study has important implications for public health. First, we emphasize the potential of task shifting to nurses to enhance healthcare access, particularly in regions with healthcare professional shortages. We also highlight the effectiveness of nurse-led chronic disease management, which supports the argument for efficient healthcare models, particularly in resource-constrained settings, where cost-effectiveness is paramount. Our research also indicates that nurse-led care can have a positive impact on patient satisfaction and outcomes, advocating for a shift towards patient-centered care models. This transformation is critical for improving management of chronic diseases. Nurse-led care also enhances continuity, ensuring that patients receive sustained support, education, and follow-up. This is of utmost importance in the effective management of chronic conditions and prevention of complications. Furthermore, our study highlighted the multifaceted role of nurses, extending beyond individual patient care to community health promotion. Nurses' involvement in addressing the risk factors for chronic diseases at the community level is crucial. Moreover, our research suggests potential avenues for the development of training programs and educational initiatives designed to equip nurses with the skills and knowledge necessary for effective chronic disease management. Furthermore, our study calls for policymakers in each country to formally define, recognize, and support nurses' roles in chronic disease management. This recognition could lead to the establishment of improved regulatory frameworks and increased resources for nurse-led care. Finally, our research contributes to data-driven decision-making within healthcare systems, enabling administrators to make informed choices regarding resource allocation.”

Conclusions:

1. Consider beginning without the phrase 'despite these limitations'.

It was done. 

2. Lines 383-384: Instead of "will guide," consider "serves as a guide for future research."

We modified it.

3. It is not customary to include citations in the conclusions section.

We modified it.

Figures and Supplementary files:

1. Fig 2: Please rephrase the caption for better clarity. Consider using a term other than "repartition" for improved adequacy.

We modified this term by “distribution”.

2. S2 Table: Consider repeating header rows for enhanced clarity.

We modified it.

Reviewer #2:

The authors have chosen a very interesting topic. The world is faced with a growing number of chronic diseases.

Thank you for your positive comment. 

• In line 123, the sentence seems incomplete, I suggest that the authors complete the sentence

We modified this sentence.

• In line 135 – well done to the authors -the review question is relevant in achieving the aim and objectives of the study.

Thank you for your positive comment. 

• In terms of the results section, I suggest that the authors should start by describing the types of the studies reviewed, then present the reviewed results. The contents in line 210-219 may have been presented before the roles and knowledge of the nurses.

We modified the order in the results section. 

• It should have been of importance if the findings from the review were presented in themes, the reader of the manuscript should know the types of roles and the knowledge found in the review.

Themes have been synthesized in the Discussion section, and we posit that pathology-specific synthesis is of paramount importance. This approach enhances the practicality and clarity of the results obtained from our research, particularly when considering the various pathologies addressed. This methodology enables the identification of interventions specific to each pathology in response to encountered issues, thus simplifying the process for the stakeholders. For instance, it is more straightforward to pinpoint mental health-related interventions in response to particular challenges than to seek general therapeutic education practices applicable across all contexts. Consequently, we have maintained the structure of the results section to reflect this comprehensive approach.

• In line 282-300, the authors presented the barriers and facilitators, whereas in the aims of the study the authors talked about challenges and gaps. This creates a confusion. It feels like it is a two in one study. I suggest the authors be consistent with the words used.

We modified the terms and standardize them.

• Line 315-324, the paragraph does not have reference, the starting of the paragraph starts with “To our knowledge …” which may show assumption, I believe there has been more reviews, this review may be adding to the available or bringing in new knowledge. I suggest that the authors look into paragraph.

We have revised the paragraph to present a more nuanced perspective, although we did not find any additional reviews on this specific topic. This section does not include references, as it serves as a synthesis of our research findings.

• The study reviewed 111 studies, however in line 343-247, only two studies were included in relation to the barriers and facilitators.

The two studies cited are not included in the review. These studies are used here to discuss the challenges and gaps identified in our review. 

Reviewer #3:

This manuscript provides a comprehensive overview to help guide the management of NCDs in Africa. There are a few recommendations. 

Thank you for your positive comment. 

In the introduction section, the authors should articulate the nursing roles for the management of NCDs as defined by nursing organizations within the country. This will enable a thorough discussion of this information with the study's findings. 

It is difficult to use country-specific definitions of nursing care. Indeed, there is not necessarily an authority regulating the nursing profession in every country, and because this is a global description, we wanted to use a universal international definition of nursing care.

Additionally, it would be advantageous to provide a clear description of the levels of evidence present in the included studies.

The quality scores of the included studies can be found in Table S3. It's important to note that, as a scoping review, the evaluation of study quality and level of evidence was undertaken to enhance the overall quality of our research and to offer readers a critical appraisal of the studies' findings. This assessment was carried out with the objective of ensuring transparency and rigor in our review process, although it is not considered a methodological mandatory requirement. 

We believe that the revisions we have implemented significantly improve the overall clarity and impact of our work. The revised manuscript now aligns more closely with the high standards set by PLOS One, and we are confident that it will contribute meaningfully to the scientific community.

Once again, we express our gratitude for the guidance provided by the reviewers and yourself. We are committed to ensuring the highest quality of our research and welcome any further suggestions for improvement.

Please find attached the revised version of the manuscript.

Thank you for your continued support, and we eagerly await your decision on our revised submission.

Sincerely yours.

---

## [Decision Letter · Decision Letter 2]

5 Dec 2023

PONE-D-23-15682R2Role and knowledge of nurses in the management of non-communicable diseases in Africa: a scoping reviewPLOS ONE

Dear Dr. Toniolo,

Thank you for submitting your manuscript to PLOS ONE. After careful consideration, we feel that it has merit but does not fully meet PLOS ONE’s publication criteria as it currently stands. Therefore, we invite you to submit a revised version of the manuscript that addresses the points raised during the review process.

We look forward to receiving your revised manuscript.

Kind regards,

Ermel Johnson, MD, MPH, PhDc

Academic Editor

PLOS ONE

Journal Requirements:

Reviewers' comments:

Reviewer's Responses to Questions

**Comments to the Author**

1. If the authors have adequately addressed your comments raised in a previous round of review and you feel that this manuscript is now acceptable for publication, you may indicate that here to bypass the “Comments to the Author” section, enter your conflict of interest statement in the “Confidential to Editor” section, and submit your "Accept" recommendation.

Reviewer #3: All comments have been addressed

Reviewer #4: All comments have been addressed

2. Is the manuscript technically sound, and do the data support the conclusions?

Reviewer #3: Partly

Reviewer #4: Yes

3. Has the statistical analysis been performed appropriately and rigorously? 

Reviewer #3: N/A

Reviewer #4: N/A

4. Have the authors made all data underlying the findings in their manuscript fully available?

Reviewer #3: Yes

Reviewer #4: Yes

5. Is the manuscript presented in an intelligible fashion and written in standard English?

Reviewer #3: Yes

Reviewer #4: Yes

6. Review Comments to the Author

Reviewer #3: In the introduction section, the authors should articulate the nursing roles for the management of NCDs as defined by nursing organizations within the region. This will be consistent with the manuscript title and facilitates a comprehensive discussion of this information with the study's findings.

Reviewer #4: Thank you for addressing all the comments. The manuscript can be accepted for publication. The publication will assist other researchers on integrated management of communicable and non communicable diseases.

7. PLOS authors have the option to publish the peer review history of their article (what does this mean?). If published, this will include your full peer review and any attached files.

Reviewer #3: No

Reviewer #4: No

---

## [Author Response · Author response to Decision Letter 2]

6 Dec 2023

Dear Ermel Johnson, 

We would like to express our gratitude to you and the esteemed reviewers for dedicating valuable time and providing insightful feedback on our manuscript PONE-D-23-15682R1 entitled “Role and knowledge of nurses in the management of non-communicable diseases in Africa: a scoping review”. 

Your constructive comments have undoubtedly enhanced the quality and rigor of our research, and we are honored by the opportunity to address these concerns and suggestions.

In this response letter, we have diligently addressed below each reviewer's comments, providing detailed explanations of the revisions we have made to the manuscript. The changes were highlighted in the manuscript. 

Journal Requirements :

It was done. We added two references linked to the reviewers comments. 

Reviewer #3:

1. In the introduction section, the authors should articulate the nursing roles for the management of NCDs as defined by nursing organizations within the region. This will be consistent with the manuscript title and facilitates a comprehensive discussion of this information with the study's findings.

We added a paragraph in the introduction, L130-145. 

Reviewer #4:

1. Thank you for addressing all the comments. The manuscript can be accepted for publication. The publication will assist other researchers on integrated management of communicable and non communicable diseases.

Thank you very much. 

We believe that the revisions we have implemented significantly improve the overall clarity and impact of our work. The revised manuscript now aligns more closely with the high standards set by PLOS One, and we are confident that it will contribute meaningfully to the scientific community.

Once again, we express our gratitude for the guidance provided by the reviewers and yourself. We are committed to ensuring the highest quality of our research and welcome any further suggestions for improvement.

Please find attached the revised version of the manuscript.

Thank you for your continued support, and we eagerly await your decision on our revised submission.

Sincerely yours.

---

## [Editor Report · Decision Letter 3]

2 Jan 2024

Role and knowledge of nurses in the management of non-communicable diseases in Africa: a scoping review

PONE-D-23-15682R3

Dear Dr. Toniolo,

We’re pleased to inform you that your manuscript has been judged scientifically suitable for publication and will be formally accepted for publication once it meets all outstanding technical requirements.

Kind regards,

Ermel Johnson, MD, MPH, PhDc

Academic Editor

PLOS ONE
---

## [Editor Report · Acceptance letter]

24 Mar 2024

PONE-D-23-15682R3 

PLOS ONE

Dear Dr. Toniolo, 

I'm pleased to inform you that your manuscript has been deemed suitable for publication in PLOS ONE. Congratulations! Your manuscript is now being handed over to our production team.

Kind regards, 

on behalf of

Dr. Ermel Johnson 

Academic Editor

PLOS ONE